# Endoribonuclease DNE1 Promotes Ethylene Response by Modulating *EBF1/2* mRNA Processing in *Arabidopsis*

**DOI:** 10.3390/ijms25042138

**Published:** 2024-02-10

**Authors:** Yan Yan, Hongwei Guo, Wenyang Li

**Affiliations:** 1Harbin Institute of Technology, Harbin 150001, China; 11849490@mail.sustech.edu.cn; 2Department of Biology, School of Life Sciences, Institute of Plant and Food Science, Southern University of Science and Technology, Shenzhen 518055, China; 3Key Laboratory of Molecular Design for Plant Cell Factory, Guangdong Higher Education Institute, School of Life Sciences, Southern University of Science and Technology, Shenzhen 518055, China

**Keywords:** plant hormone ethylene, EBF1/2, Endoribonuclease DNE1, EIN2, mRNA processing

## Abstract

The gaseous phytohormone ethylene plays a crucial role in plant growth, development, and stress responses. In the ethylene signal transduction cascade, the F-box proteins EIN3-BINDING F-BOX 1 (EBF1) and EBF2 are identified as key negative regulators governing ethylene sensitivity. The translation and processing of *EBF1/2* mRNAs are tightly controlled, and their 3′ untranslated regions (UTRs) are critical in these regulations. However, despite their significance, the exact mechanisms modulating the processing of *EBF1/2* mRNAs remain poorly understood. In this work, we identified the gene *DCP1-ASSOCIATED NYN ENDORIBONUCLEASE 1* (*DNE1)*, which encodes an endoribonuclease and is induced by ethylene treatment, as a positive regulator of ethylene response. The loss of function mutant *dne1-2* showed mild ethylene insensitivity, highlighting the importance of DNE1 in ethylene signaling. We also found that DNE1 colocalizes with ETHYLENE INSENSITIVE 2 (EIN2), the core factor manipulating the translation of *EBF1/2*, and targets the P-body in response to ethylene. Further analysis revealed that DNE1 negatively regulates the abundance of *EBF1/2* mRNAs by recognizing and cleaving their 3′UTRs, and it also represses their translation. Moreover, the *dne1* mutant displays hypersensitivity to 1,4-dithiothreitol (DTT)-induced ER stress and oxidative stress, indicating the function of DNE1 in stress responses. This study sheds light on the essential role of DNE1 as a modulator of ethylene signaling through regulation of *EBF1/2* mRNA processing. Our findings contribute to the understanding of the intricate regulatory process of ethylene signaling and provide insights into the significance of ribonuclease in stress responses.

## 1. Introduction

The gaseous phytohormone ethylene exerts a profound impact on the whole life cycle of plants, particularly in response to a wide range of stresses such as cold, heat, drought, flooding, salinity, and diseases [1]. Researchers have extensively investigated the mechanism of ethylene actions in the reference plant *Arabidopsis* over the past two decades, utilizing genetic and molecular biological methods to identify a series of ethylene mutants and subsequently establish a relative linear ethylene signal transduction pathway [2,3,4]. The most upstream sensor responsible for ethylene perception in this pathway is the ethylene receptors, which comprise five proteins belonging to the same family: ETHYLENE RESPONSE 1 (ETR1), ETHYLENE RESPONSE SENSOR 1 (ERS1), ETR2, ERS2, and ETHYLENE INSENSITIVE 4 (EIN4) [5,6]. In the absence of ethylene, the endoplasmic reticulum (ER) membrane-located receptors interact with and activate the downstream factor Raf protein kinase CONSTITUTIVE TRIPLE RESPONSE 1 (CTR1) [7]. CTR1 then phosphorylates another ER-localized protein, EIN2, at Ser645 and Ser924, resulting in the prompt turnover of EIN2 via two F-box proteins, EIN2 TARGETING PROTEIN 1 (ETP1) and ETP2, thus preventing ethylene response [8,9,10].

EIN2 is the central positive regulator in ethylene signaling cascade that modulates almost every aspect of ethylene responses, as evidenced by the fact that *ein2* loss-of-function mutants are completely insensitive to ethylene [8]. While the function of the putative membrane-spanning domain at amino terminus remains unclear, the hydrophilic C-terminal end of EIN2 (CEND) is shown to relay signaling from ER to the nucleus. Upon exposure to ethylene, the binding of ethylene molecules to the receptors with the help of monovalent copper ions (Cu+) leads to the inactivation of the receptor-CTR1 complex. This subsequently causes the unphosphorylation of CEND, resulting in its release from the rest of EIN2 through proteolytic cleavage, which activates downstream ethylene signaling [10,11,12]. Immediately after EIN2 are two nucleus-located transcription factors, EIN3 and its homolog EIN3-LIKE 1 (EIL1), which modulate the expression of the majority of ethylene-responsive genes [13]. Notably, *EIN3* and *EIL1* are not transcriptionally regulated by ethylene. Instead, the stability of EIN3/EIL1 proteins is modulated by the 26S ubiquitin–proteasome system in an EIN3-BINDING F-BOX 1 (EBF1) and EBF2-dependent manner in the absence of ethylene [14,15,16]. Recent studies have revealed that EIN2 promotes ethylene response through at least three ways: (i) CEND directly inhibits the translation of *EBF1/2* mRNAs by targeting them to P-body in the cytoplasm to enhance the stabilization of EIN3/EIL1 [17,18]; (ii) EIN2 triggers the accumulation of EIN3/EIL1 through inducing the proteasomal degradation of EBF1/2 proteins [19]; (iii) the cleaved CEND translocates into the nucleus, where it recruits the histone-binding protein ENAP1 (EIN2 NUCLEAR-ASSOCIATED PROTEIN 1) to modify the histone acetylation of EIN3/EIL1-targeting genes [13].

As sessile organisms, plants have developed adaptation mechanisms to tolerate biotic and abiotic stresses. To establish optimal responses to these stimuli, extensive regulation at both transcriptional and post-transcriptional levels is required. EBF1 and EBF2, as vital repressors controlling ethylene sensitivity, are subject to elaborate regulation by different mechanisms. The RING-type E3 ligase SDIR1 modulates ethylene signaling and plant adaptation to temperature changes by targeting EBF1/EBF2 for degradation and promoting EIN3 accumulation [20]. EIN3/EIL1 directly bind to the promoters of the *EBF1/2* genes, thereby activating their transcription and forming a negative feedback loop [21,22]. Additionally, EIN5, also known as EXORIBONUCLEASE4 (XRN4) and homologous with yeast XRN1, is involved in the degradation of 5′-decapped mRNA as well as mRNA fragments arising from internal cleavage events such as small RNA-mediated slicing. Previous studies have found that loss of function of *EIN5* results in overaccumulation of *EBF1/2* mRNAs and leads to an ethylene-insensitive phenotype [23,24]. Interestingly, while the turnover of *EBF1/2* mRNAs is not affected by the mutation of *EIN5*, it is striking that the 3′ end transcripts, which arise from cleavage of 3′ untranslated regions (UTRs) of *EBF1* and *EBF2* mRNAs, are over-accumulated in the *ein5* mutant [25]. Subsequent studies discovered that EIN2, together with EIN5, nonsense-mediated decay (NMD) proteins UPF1/2/3, and poly (A)-binding proteins PAB2/4/8, associates with the 3′UTRs of *EBF1*/*2* mRNAs to repress their translation in the presence of ethylene [17,18]. Furthermore, the 3′UTRs of *EBF1* and *EBF2* mRNAs are sufficient to confer translational regulation and are required for the proper activation of ethylene response. It is intriguing to find that exogenous overexpression of *EBF1* or *EBF2 3*′*UTR* in wild-type *Arabidopsis* leads to an ethylene-insensitive phenotype. The overexpressed *3*′*UTR* transcript acts as a “sponge” to absorb the so-called translation inhibitors, which supposedly bind to 3′UTRs of endogenous *EBF1/2* mRNAs, thus relieving the repression on *EBF1/2* mRNAs [17]. However, despite the significance of *EBF1/2* 3′UTRs, the enzymatic processing of shorter RNA fragments at the 3′ ends from full-length *EBF1/2* transcripts is not well-understood.

Dedicated RNA processing systems are essential machines for cells to control the levels of specific transcripts. Eukaryotes contain several distinctive systems that mediate RNA processing, such as exo- and endo-ribonuclease (RNase), exosome, spliceosome, NMD system, decapping complex, and RNA-induced silencing complex (RISC). Over the past 20 years, there has been intense attention given to the NMD pathway, as it plays an important role in maintaining the quality of mRNA by selectively degrading aberrant transcripts or inhibiting their translation [26]. The NMD machinery targets mRNAs containing abnormal termination codons (TCs), upstream open reading frames (uORFs), long 3′ UTRs, and introns located downstream of termination codons [26,27,28]. After being activated by phosphorylation and associating with UPF2 and UPF3, the core factor UPF1 exerts its helicase activity and recruits a protein complex composed of SMG5, SMG7, and SMG6 to ultimately degrade substrate mRNAs or prevent their translation [26,29]. In animals, the endoRNase SMG6 cleaves NMD targets, and the resultant mRNA fragments are further degraded through exonucleolytic activity [29,30]. Yet, the RNA endonuclease in plant NMD pathway has not been identified. Recently, an NYN domain-containing endoribonuclease DCP1-ASSOCIATED NYN ENDORIBONUCLEASE 1 (DNE1) was found to directly interact with DCP1 and UPF1 and co-localizes with them in P-body to modulate mRNA stability and translation [31,32]. Although DNE1 is a cytoplasmic endoRNase, degradome sequencing revealed that the majority of DNE1 substrates are not NMD targets [33]. Additionally, previous research has shown that UPF1/2/3 and DCP1 are involved in the EIN2-mediated translational inhibition of *EBF1/2* [17,18]. However, the involvement of DNE1 in the ethylene signal transduction cascade has not yet been determined.

In this work, we discovered that *DNE1*, encoding an endoribonuclease, is induced by ethylene treatment and positively regulates ethylene response. The loss of function mutant *dne1* shows a mild ethylene-insensitive phenotype, and the *ein5 dne1* double mutant displays a significant ethylene-insensitive phenotype. Our results unveiled that DNE1 colocalizes with EIN2 and targets P-body in response to ethylene. Further analysis revealed that DNE1 negatively regulates the abundance of *EBF1/2* mRNAs by recognizing and cutting their 3′UTRs. In addition, we showed that the *dne1* mutant is hypersensitive to 1, 4-dithiothreitol (DTT)-induced ER stress and oxidative stress. Our study thus illustrates that DNE1 acts as an essential modulator of ethylene signaling through regulating the processing of *EBF1/2* mRNAs. Our research uncovers new roles of DNE1 in the plants’ response to abiotic stress and provides novel insights into the biological functions of mRNA post-transcriptional regulation.

## 2. Results

### 2.1. DNE1 Positively Regulates Ethylene Response in Arabidopsis

Previous studies demonstrated that UPF1 interacted with the *EBF1/2* 3′UTR and UPF1 was responsible for the translation inhibition of *EBF1/2* [17,18]. When translation is blocked, mRNA is likely to be recognized as an aberrant transcript by the RNA quality control machinery, and consequently degraded by RNases [34,35,36]. Given that DNE1 interacted with UPF1 [32], we proposed that DNE1 is involved in the ethylene signaling pathway. Firstly, we analyzed the expression pattern of *DNE1* upon ethylene treatment by searching for *DNE1* mRNA levels in a public database for *Arabidopsis* ethylene treatment transcriptome (accessed on 1 January 2024 from EIN3-Ethylene Chip-seq/RNA-seq browser, http://neomorph.salk.edu/dev/pages/EIN3.html). The results showed that *DNE1* expression levels were significantly increased in etiolated seedlings after ethylene treatment (Figure 1A), suggesting that *DNE1* is upregulated by ethylene, implying that DNE1 is involved in the ethylene response.

To gain insight into the role of DNE1 in the ethylene signaling pathway, we identified a T-DNA insertion line disrupting the DNE1 coding sequence, referred to as *dne1-2* (different from the *dne1-1* allele in [32,33]). Next, we wanted to investigate the ethylene response phenotype of the *dne1-2* mutant. The triple response is an essential characteristic of the ethylene response phenotype, with the shortening of hypocotyl and root lengths as well as a decreased apical hook [37]. The compound 1-aminocyclopropane-1-carboxylic acid (ACC) is a precursor to ethylene, which is oxidized by the ACC oxidase (ACO) to produce ethylene [38]. We tested the triple response phenotypes of Col-0, *ein5-1*, *dne1-2*, *ein5-1 dne1-2*, *upf1-10*, and *ein5-1 upf1-10* mutants grown on MS medium supplied with or without ACC in the dark for three days. As shown in Figure 1B–D, when grown on MS medium without ACC, the hypocotyl lengths of these seedlings showed no obvious difference. When treated with ACC, Col-0 displayed a short hypocotyl, short root, and exaggerated apical hook, while *ein5-1* was insensitive to ACC, as it exhibited longer hypocotyl and root lengths as well as a mild apical hook. The *dne1-2* mutant showed slightly ethylene-insensitive phenotype compared with Col-0, while the *ein5-1 dne1-2* double mutant displayed an enhanced ethylene-insensitive phenotype than the *ein5-1* single mutant. Similarly, the *upf1-10* single mutant showed a mild ethylene-insensitive phenotype, and the *ein5-1 upf1-10* double mutant displayed an enhanced ethylene-insensitive phenotype compared to the single mutant (Figure 1B–D). These data reveal that DNE1 positively regulates the ethylene response.

### 2.2. DNE1 Co-Localizes with EIN2 in P-Body in Response to Ethylene

Membrane-less organelles, including P-bodies and stress granules, are usually formed in plant cells in response to a variety of environmental stresses, such as heat shock, low oxygen levels (hypoxia), salinity, and hyperosmolarity [39,40,41]. When plants are exposed to ethylene, EIN2 associates with P-body components, including EIN5, UPF1, and PABPs, and targets *EBF1/2* mRNA to P-body to subsequently inhibit the translation of *EBF1/2* [17,18]. DNE1, which has been found to directly interact with UPF1 and DCP1 [31,32], is assumed to be present in P-body and related to EIN2 upon ethylene treatment. Therefore, we examined the subcellular allocation of DNE1, as well as its connection with EIN2 and P-body components. Here, we transiently co-expressed DNE1-YFP with EIN2 CEND-CFP, DCP2-CFP, UPF1-CFP, or EIN5-CFP into tobacco leaves. Upon ACC treatment, EIN2 CEND was induced to form cytoplasmic foci in P-body [17]. DNE1 was observed to co-localize with EIN2 CEND in cytoplasmic foci (Figure 2A). Although DNE1 only interacts with DCP1 but not DCP2 [32], DCP2, which is localized in P-body and interacts with DCP1 [42], showed co-localization with DNE1 (Figure 2B). In order to verify the identity of these condensates, we determined the subcellular localization of DNE1 and found co-localization between DNE1 and UPF1, and EIN5 (Figure 2B–D). This is consistent with the results of previous studies indicating that DNE1 was localized in P-body [32,33]. Minor signals of DNE1 were also observed in the nucleus, which is consistent with previous research, as the cytoplasmic membrane systems are connected to the nuclear envelope [33]. In conclusion, we found that DNE1 co-localizes with EIN2 and P-body components upon ethylene treatment, implying that DNE1 participates in ethylene signaling regulation.

### 2.3. DNE1 Impairs the Expression of EBF1/2 3′UTR-Tailed Reporter Genes

Previous research revealed that *EBF1/2* mRNA moved to P-bodies upon ethylene treatment [17,18]. The cytoplasmic 5′-3′ exoribonuclease EIN5 was reported to modulate the levels of *EBF1/2* mRNA, and small RNA fragments corresponding to the 3′UTRs of *EBF1/2* mRNAs are over-accumulated in the *ein5* mutant [23,24,25], suggesting that *EBF1/2* mRNA might undergo endoribonucleic cleavage. As DNE1 is an active NYN domain endonuclease [32,33], we wondered if DNE1 altered the stability of *EBF1/2* mRNA. The MS2 system, based on the principle of MS2 protein binding to RNA with a matching binding site, is a powerful tool for visualizing and quantifying RNA molecules within living cells and has been utilized in various studies, including those investigating mRNA transport, localization, and regulation of gene expression [43]. Here, we conducted the MS2 system to study the effect of DNE1 on *EBF1/2* mRNA. We fused six tandem repeats of MS2 binding sites (6×MS2 binding site, referred to as 6×MS2bs) with either mRFP (mRFP-6×MS2bs, called RM) or *EBF1/2* 3′UTR (mRFP-6×MS2bs-*EBF1/2* 3′UTR, RM1U/RM2U for short) to produce reporter RNAs (Figure 3A). The MS2 coat protein (referred to as MS2CP) was fused with either YFP-Flag (MS2CP-YFP) or DNE1-YFP-Flag (MS2CP-DNE1-YFP) to generate effector proteins (Figure 3A). Due to the high affinity of MS2CP for the MS2bs stem-loop structure, MS2CP-tagged effector proteins will be tethered to RNA sequences containing the 6×MS2bs element (Figure 3B). In a transient assay, the reporter RNAs and effector proteins were co-expressed in tobacco leaves to investigate DNE1’s endoribonucleic activity by monitoring mRFP fluorescence density (Figure 3C). We then found that MS2CP-DNE1-YFP significantly inhibited the expression of RM, RM1U, and RM2U compared to MS2CP-YFP, as the fluorescence of mRFP was much lower in the cells expressing MS2CP-DNE1-YFP than in those expressing MS2CP-YFP, indicating the suppressive effect of DNE1 on tethered RNA (Figure 3D–F). Additionally, we found that the expression of RM1U and RM2U was significantly reduced by DNE1-YFP when compared to RM, as indicated by their lower mRFP fluorescence (Figure 3D–F). This demonstrates that the inhibitory effect of DNE1 on the *EBF1/2* 3′UTR is not dependent on its connection to the MS2 binding site.

Next, we tested the mRNA level of *EBF1/2* 3′UTR and *mRFP* by qRT–PCR. Since the *Kanamycin* gene is driven by another 35S promoter in the same vector expressing RM/RM1U/RM2U, we used *Kanamycin* as the reference gene of qRT–PCR. As shown in Figure 3G, when RM was co-expressed with either MS2CP-YFP or DNE1-YFP, the expression levels of *mRFP* were comparable. Meanwhile, the abundance of *mRFP* was largely decreased when RM was co-expressed with MS2CP-DNE1-YFP. We also uncovered that the co-expression of RM1U/RM2U with MS2CP-DNE1-YFP led to a significant decrease in the abundances of *mRFP* compared to the combinations of RM1U/RM2U with MS2CP-YFP (Figure 3H). These results indicated that DNE1 was capable of suppressing the expression of genes tethered to it. Furthermore, when RM1U/RM2U was co-expressed with DNE1-YFP instead of MS2CP-YFP, the expression levels of *mRFP* were significantly reduced (Figure 3I). In line with the observations in Figure 3D–F, this finding suggests that DNE1 selectively suppresses reporter RNA expression levels by recognizing *EBF1* and *EBF2* 3′UTRs.

### 2.4. DNE1 Is Required for 3′ End Cleavage of EBF1/2 mRNA

In order to verify the repressive effect of DNE1 on *EBF1/2* mRNA, we detected mRNA levels of *EBF1/2* in etiolated seedlings of Col-0, *ein5-1*, *dne1-2*, and *ein5-1 dne1-2* grown on MS medium supplemented with and without 10 μM ACC. As shown in Figure 4A,B, *EBF1/2* mRNA levels are higher in *ein5-1* compared with Col-0, consistent with previous research [23,24]. *EBF1/2* mRNA levels were slightly higher in *dne1-2* than in Col-0. Meanwhile, *EBF1/2* mRNA levels were significantly higher in *ein5-1 dne1-2* than in *ein5-1* (Figure 4A,B). These data are in accordance with the triple response phenotype in Figure 1B.

We next sought to identify 3′ fragments of *EBF1/2* that could be generated by DNE1. We used 5′-RNA Ligase-Mediated Rapid Amplification of cDNA Ends (5′-RLM-RACE) to identify 3′ fragments of *EBF1/2* in etiolated seedlings of Col-0, *ein5-1*, *dne1-2*, and *ein5-1 dne1-2* grown on MS medium supplemented with 10 μM ACC. For *EBF1*, we detected *EBF1* 3′ fragments in *ein5-1* and *ein5-1 dne1-2* but not in Col-0 and *dne1-2*, and the abundance of *EBF1* 3′ fragments in *ein5-1 dne1-2* was significantly decreased than that in the *ein5-1* mutant (Figure 4C). It is suggested that DNE1 is required for *EBF1* mRNA cleavage. For *EBF2*, we detected *EBF2* 3′ fragments in all genotypes, and the abundance of *EBF2* 3′ fragments in *ein5-1 dne1-2* was much lower than that in *ein5-1* (Figure 4C). Taking into consideration the fact that the abundance of *EBF2* 3′ fragments in Col-0, *ein5-1*, and *dne1-2* showed comparable levels (Figure 4C), we infer that *EBF2* mRNA 3′UTR is subject to more complicated regulation. In conclusion, these data suggest that DNE1 selectively cleaves *EBF1/2* 3′UTR, but *EBF1* mRNA is a preferred target.

As we failed to create specific antibodies against EBF1 and EBF2 proteins, next, we analyzed the protein level of EIN3 protein in the seedlings mentioned in Figure 4A. Consistent with the ethylene-insensitive phenotypes, the EIN3 protein levels were significantly decreased in *ein5-1*, *dne1-2*, and *ein5-1 dne1-2* double mutant under the treatment of ACC (Figure 4D). This result implies that the corresponding protein levels of EBF1/2 might be up-regulated when EIN5 or DNE1 is mutated. Although the mRNA levels of *EBF1/2* in *ein5-1* and *ein5-1 dne1-2*, with or without ACC treatment, were identical, the protein abundance of EIN3 was decreased further in *ein5-1 dne1-2* than in *ein5-1* (Figure 4D). This result indicates that the translation of *EBF1/2* mRNA might be up-regulated when EIN5 and DNE1 simultaneously lose their activities. Taking together, these findings suggested that DNE1 mediates the cleavage of *EBF1/2* mRNA 3′ ends and takes part in repressing the translation of *EBF1/2* mRNAs together with EIN5 and other factors.

### 2.5. DNE1 Positively Regulates Arabidopsis Resistance to ER Stress and Oxidative Stress

In plants, there are various types of RNases (ribonucleases) that play crucial roles in the processing and degradation of RNA, which are important for the regulation of plant growth and development, as well as the response to stress conditions [44,45,46]. The *dne1* mutant showed hypersensitivity to osmotic stress [47], and the DNE1 overexpression line exhibited more tolerance to oxidative stress [48]. To better elucidate the biological roles of DNE1, we treated both wild-type and *dne1-2* mutant plants with DTT (1,4-dithiothreitol) and methyl viologen (MV). DTT can disrupt the redox environment of the endoplasmic reticulum by interfering with the formation of disulfide bonds, thus causing endoplasmic reticulum stress [49,50]. The *dne1-2* mutant showed hypersensitivity to DTT-induced ER stress with smaller leaves and lower fresh weight than Col-0 (Figure 5A,B). On the other hand, the *dne1-2* mutant displayed hypersensitivity to MV-induced oxidative stress with shorter root lengths and a higher bleaching rate (Figure 5C–E), confirming the previously published data [47,48]. Overall, we found that DNE1 is involved in ER stress and oxidative stress, providing new insights for a comprehensive understanding of the biological roles of DNE1.

## 3. Discussion

The plant hormone ethylene plays vital roles in plant growth, development, and stress responses. Uncovering the mode of action of ethylene will facilitate our efforts to improve crop plants yield, quality, resilience, and adaptations to environmental cues [4,20,51,52]. A linear ethylene signaling pathway has been established in *Arabidopsis* two decades ago, where ethylene is perceived by ER membrane-located receptors. This signal is then relayed and transduced to the nucleus by EIN2. Upon ethylene treatment, EIN2 inhibits the translation of *EBF1/2* mRNA [17,18]. Hence, the decrease in EBF1/2 protein levels leads to an increase in EIN3/EIL1 transcription factors, resulting in the induction of the expression of ethylene-responsive genes [13]. While the regulatory mechanisms of key components in the ethylene signaling cascade, including CTR1, EIN2, EIN3/EIL1, are extensively studied, the details of how RNA molecules are involved in and regulate ethylene response remain unclear. This aspect of the pathway, concerning RNA processing, is still an area requiring further research and elucidation. Compared to RNA research in animals, many RNA technologies and methods are not widely applied in plant research. This might be the reason why there are less investigations in this area.

Taken together, the 3′UTRs of *EBF1/2* transcripts recruit RNA processing factors including EIN2, EIN5, UPF1, and DNE1 to manipulate the abundance and translation of *EBF1/2* mRNAs. DNE1, acting as a nucleic acid endonuclease, recognizes and cleaves the 3′UTRs of *EBF1/2* mRNAs, thereby modulating the ethylene response through negatively regulating the abundance as well as the translation of *EBF1/2* (Figure 6A). EBF1/2 proteins negatively regulate the key transcription factors EIN3 and EIL1; hence, DNE1 promotes the plant response to ethylene. In the *ein5-1* mutant, the abundance of *EBF1/2* mRNA is slightly higher than that in the wild type. DNE1 promotes the accumulation of 3′UTR fragments via its endoribonucleic activity. The over-accumulated 3′UTR fragments subsequently function as a molecular sponge to mitigate EIN2-mediated translation repression of *EBF1/2* mRNAs (Figure 6B). In the *ein5-1 dne1-2* double mutant, the abundance of *EBF1/2* mRNA is higher than that in both wild-type Col-0 and the *ein5-1* mutant. Meanwhile, the translational inhibition of *EBF1/2* mRNA is further released. These allow EBF1/2 proteins to increase to a higher level and result in a more pronounced ethylene-insensitive phenotype (Figure 6C). Our findings are helpful for obtaining a deeper understanding of the molecular mechanisms of the ethylene signaling pathway in plants.

In the ethylene signaling pathway, genetic evidence indicates that EIN5 is positioned at the downstream of CTR1 and upstream of EBF1/2, and that the levels of *EBF1/2* mRNA are significantly upregulated in the *ein5* mutant [23,24]. Additionally, shorter RNA species corresponding to the 3′UTR of *EBF1/2* mRNA accumulate to a higher level in the *ein5* mutant [23,24]. The mRNA half-life measurement indicates that the half-life of the full-length *EBF1/2* mRNA does not differ much between Col-0 and the *ein5* mutant, but the half-life of the 3′UTR fragment is significantly extended in the *ein5* mutant [25]. This suggests that EIN5 primarily degrades the small RNA fragments derived from the *EBF1/2* 3′UTR rather than the full-length *EBF1/2* mRNA. However, the molecular mechanism behinds the processing of *EBF1/2* mRNA into 3’ fragments is not well-understood.

In this study, we found that DNE1, an active endoribonuclease containing the NYN domain, could cleave the *EBF1/2* mRNA to generate 3′ end fragments. Although the *dne1-2* mutants did not display a very strong ethylene-insensitive phenotype, the *ein5-1 dne1-2* double mutant displayed a stronger ethylene-insensitive phenotype than both *ein5-1* and *den1-2* (Figure 1B). We noticed that the abundance of *EBF1* 3′ fragments decreased to about 30% while the abundance of *EBF2* 3′fragments decreased to about 50% in *ein5-1 dne1-2* compared with *ein5-1* (Figure 4C). These are the possible reasons to explain the weak phenotype of *dne1-2*: (1) there might be other functionally redundant RNases involved in the cleavage of *EBF1/2* mRNA, and this is in agreement with the fact that diversified NYN- and PIN-domain ribonucleases could be found in the *Arabidopsis* genome; (2) the lack of DNE1 protein slightly dampens but does not totally destroy the integrity and functions of EIN2-associating P-bodies; this is favored by the phenomenon that the ethylene insensitivities are gradually elevated in *ein5*, *ein5 upf1*, *ein5 upf1 pab2 pab8* mutants [17,18].

Previously, researchers have characterized DNE1 substrates by RNA degradome analysis [33]. It is noteworthy that *EBF1/2* are not identified among the substrates of DNE1 in these analyses. This may be attributed to various factors, including method-constrained experiment resolution, material-determined target abundance, and procedure-introduced randomness. *EBF1* and *EBF2* are genes that are induced dramatically by stress response, causing their transcripts and 3’ UTR fragments to remain at a relatively low level under normal conditions. Specifically, the authors employed a selection criterion of *xrn4/dne1 xrn4* log_2_FC ≥ 2, focusing on filtered XRN4-sensitive 5’P sites. It is possible that the 3′ fragments of *EBF1/2* did not align with this stringent selection threshold, leading to their exclusion from the identified substrates. Moreover, the plant materials in our work differed from those in the abovementioned study. We utilized etiolated seedlings treated with ethylene, while the prior study employed untreated green seedlings. Whether *EBF1/2* mRNAs are natural substrates of the NMD pathway is uncertain. This is because most of the substrates of DNE1 do not overlap with those of the NMD pathway. There are many enzymes responsible for RNA degradation in the cytoplasm of plant cells, such as RNases from the NYN and PIN families [32]. These enzymes might have functional redundancy. The other RNase(s) involved in the processing of *EBF1/2* mRNAs remain to be further explored.

In conclusion, we have identified that DNE1 is responsible for *EBF1/2* 3′UTR cleavage, and it also positively regulates ethylene response through targeting to P-body. Our work is important for understanding RNA processing in the ethylene signaling pathway. Additionally, it provides novel insights into understanding the biological function of DNE1.

## 4. Materials and Methods

### 4.1. Plant Material

The *Arabidopsis* mutant materials used in this study are all Col-0 ecotypes. Seeds of *dne1-2* (SALK_009495) were obtained from the Arabidopsis Biological Resource Center (ABRC) (https://abrc.osu.edu/, accessed on 10 April 2018). The *ein5-1* is preserved by our lab, which harbors a 1 bp deletion at position 4292 in exon 15. The *upf1-10* mutant is a gift from the Alonso–Stepanova Laboratory [18]. Primers used for genotype identification are listed in Appendix A.

### 4.2. Growth Condition

Surface sterilization of *Arabidopsis* seeds is conducted using an ethanol method. Seeds are washed with a sterilization solution (75% ethanol + 0.05% Triton-X surfactant) for 10 min, briefly centrifuged, and then, the sterilization solution is discarded. This is followed by two washes with 100% ethanol, each for 1 min. After vacuum-drying the seeds, they are sown on MS solid medium. After three days in the refrigerator at 4 °C, the plates are put in a light incubator. The plates are first placed in a light incubator for about 4 h. Following this incubation, the plates are wrapped with aluminum foil. These wrapped plates are then cultivated in the greenhouse for three days. For green seedlings, upon removal from the 4°C refrigerator, the green seedlings are immediately placed in the greenhouse. They remain there for a period of 12 days for their cultivation. The growth conditions for the plants used in this experiment are long-day conditions (16 h light/8 h dark) with a temperature around 22 °C and a relative humidity of 60%.

### 4.3. Gene Expression Analysis by Quantitative PCR

Total RNA from plants is extracted using the Eastep Super Total RNA Extraction Kit (Promega, Madison, WI, USA), according to the manufacturer’s instructions provided with the kit. Reverse transcription and real-time PCR are previously described [17]. Primers used for Quantitative PCR are listed in Appendix A.

### 4.4. 5′-RLM-RACE

After total RNA extraction, a GeneRacer RNA Oligo is ligated to RNAs that expose a phosphate group at the 5′ end. The RNA ligation steps are as follows. The GeneRacer™ RNA Oligo (0.25 µg) is added to 7 µL of purified total RNA (3 µg). After that, the mixture is heated at 65 °C for 5 min, then chilled on ice for 2 min. To this cooled mixture, 1 µL of 10X Ligase Buffer, 1 µL of 10 mM ATP, 1 µL of RNaseOut™ (40 U/µL, Thermo Fisher Scientific, Waltham, MA, USA), and 1 µL of T4 RNA ligase (5 U/µL, Thermo Fisher Scientific, Waltham, MA, USA) are added. The solution is then incubated at 37 °C for 1 h. After incubation, it is stored on ice for later use. Next, RNA is precipitated according to the following steps. The ligation mixture is mixed with 90 µL of DEPC water and 100 µL of phenol-chloroform. This mixture is vortexed for 30 s to ensure thorough mixing. The mixture is then centrifuged at room temperature at 13,000× *g* for 5 min. The aqueous phase is carefully transferred to a new tube after centrifugation. 2 µL of 10 mg/mL mussel glycogen (Roche, Basel, Switzerland) and 10 µL of 3 M sodium acetate (pH 5.2) are added to the aqueous phase. The mixture is mixed well. Next, 220 µL of 95% ethanol is added and the tube is vortexed again. This mixture is then frozen at −80 °C for 10 min. A centrifugation at 13,000× *g* at 4 °C for 15 min follows. The supernatant is removed post-centrifugation. The RNA pellet is washed by adding 500 µL of 70% ethanol and vortexing, then followed by another round of centrifugation at 13,000× *g* at 4 °C for 2 min. This washing step is repeated once more. Finally, the ethanol is carefully removed, and the RNA pellet is left to dry and dissolve in 10 µL of RNase-free water. Then, the RNAs are reverse transcribed using Oligo(d)T as primer. After PCR using a forward primer matching to adapter coupled with the gene-specific reverse primer, *EBF1/2* mRNA 3′ fragments are amplified and separated by agarose electrophoresis. Primers used for 5′-RLM-RACE are listed in Appendix A.

### 4.5. Vector Construction

For co-localization, the DNE1 coding sequence was amplified from *Arabidopsis* cDNA, then inserted into the vector pEGAD-GFP using EcoRI and BamHI sites, to obtain the PEGAD-DNE1 plasmids. Plasmids for EIN2-CEND-CFP, UPF1-CFP, DCP2-CFP, and EIN5-CFP were used in [17]. For tethering assays, DNE1-YFP and YPF were fused with or without MS2 coat protein to the PQG110 vector using BamHI and KpnI sites. The mRFP was fused with or without *EBF1/2* 3′UTRs to the PQG110 vector using BamHI and KpnI sites. Primers used for vector construction are listed in Appendix A.

### 4.6. Transient Expression in Nicotiana benthamiana

After approximately 4 weeks of growth, the tobacco leaves are ready for infiltration. The Agrobacterium culture is grown in liquid LB medium until an OD_600_ of about 1.0 is reached. It is then centrifuged at 4 °C, 4000 rpm for 15 min, and the supernatant is discarded. The bacteria are resuspended in tobacco leaf infiltration buffer, and then centrifuged again at 4 °C, 4000 rpm for 15 min, discarding the supernatant. The pellet is resuspended in 10 mL of tobacco leaf infiltration buffer, and the OD_600_ is measured. An appropriate volume of the culture is then diluted to an OD_600_ of 0.5. Using a 1 mL syringe, the bacterial suspension is gently injected against the underside of the tobacco leaf. Afterward, the infiltrated leaves are marked, and the plants are moved to an area with low light intensity to recover for 12 h before being placed back under normal growth conditions. Leaves can be harvested for fluorescence observation and protein extraction experiments 48 h post-infiltration.

### 4.7. Fluorescence Observation

Fluorescence observation is imaged using a Zeiss LSM880 confocal microscope (Carl Z, Oberkochen, Germany). For ACC treatment, ACC is added to the infiltration buffer to a final concentration of 10 μM/L.

### 4.8. Total Protein Extraction and Western Blot

*Arabidopsis* etiolated seedlings are collected and immediately frozen in liquid nitrogen. While still frozen, the samples are ground to a fine powder in liquid nitrogen. A half volume of protein extraction buffer (4× Bolt™ LDS Sample Buffer, Thermo Fisher Scientific, Waltham, MA, USA) is then added to the powdered sample. The mixture is shaken vigorously for thorough mixing followed by incubation at 65 °C for 10 min. After incubation, the samples are centrifuged at 22 °C at 13,000 rpm for 10 min. The protein samples, along with a protein marker, are then loaded into the wells of an SDS-PAGE gel. Electrophoresis is initiated at 80 V until the protein bands reach the interface between the stacking and resolving gel. The voltage is then increased to 130 V. Electrophoresis continues until the red band of the protein marker is near the bottom of the gel. After electrophoresis, proteins were electroblotted onto a PVDF membrane. Then, the membrane is incubated in TBST containing milk at room temperature for 1 h. The primary antibodies are diluted at 1:5000 for endogenous EIN3 protein [13] and 1:200,000 for HSP90 (Beijing Protein Innovation, Beijing, China), respectively. For secondary antibody incubation, the membrane is washed thrice with TBST for 10 min each, then incubated with the secondary antibody in TBST at room temperature for 50 min. Mouse or Rabbit anti-HRP secondary antibodies (Promega, Madison, WI, USA) are diluted at 1:10,000. After secondary antibody incubation, the membrane is washed again in TBST, then detected by a chemiluminescence reaction.

## Figures and Tables

**Figure 1 ijms-25-02138-f001:**
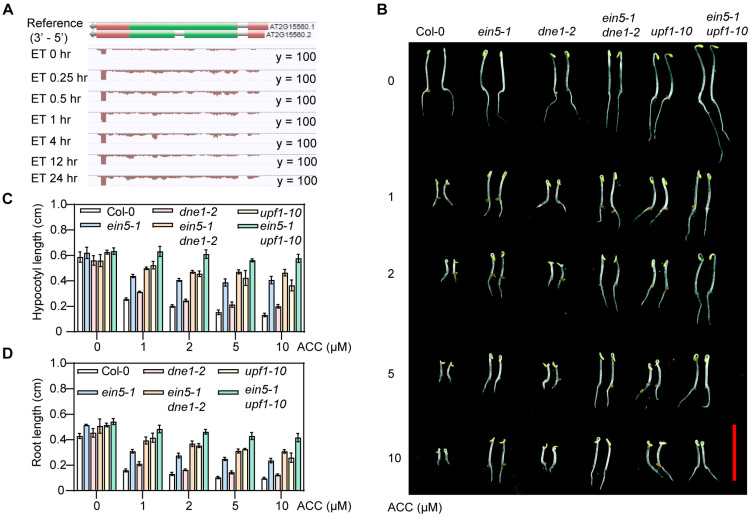
DNE1 positively regulates ethylene response in *Arabidopsis*. (**A**) The expression levels of *DNE1* in etiolated wild-type Col-0 seedlings that were treated with ethylene gas (ET) at 10 μL/L for 0, 0.25, 0.5, 1, 4, 12, and 24 h. Illustration by integrated genome view (IGV) picture accessed on 1 January 2024 from EIN3 -Ethylene Chip-seq/RNA-seq browser, http://neomorph.salk.edu/dev/pages/EIN3.html), green box represents coding sequence (CDS), red box denotes untranslated region (UTR), gray line with arrowhead stands for gene direction. (**B**) Phenotypes of Col-0, *ein5-1*, *dne1-2*, *ein5-1 dne1-2*, *upf1-10*, *ein5-1*, and *upf1-10* with or without 1-aminocyclopropane-1-carboxylic acid (ACC) treatment. Seedlings were grown on MS medium supplemented with 0, 1, 2, 5, 10 μM ACC at 22 °C for three days. Scale bar = 1 cm. (**C**,**D**) Quantification of hypocotyl lengths and root lengths of seedlings in (**B**). About 30 seedlings for each biological replicate were used, and three independent biological replicates were performed. Values are means marked with ±SD.

**Figure 2 ijms-25-02138-f002:**
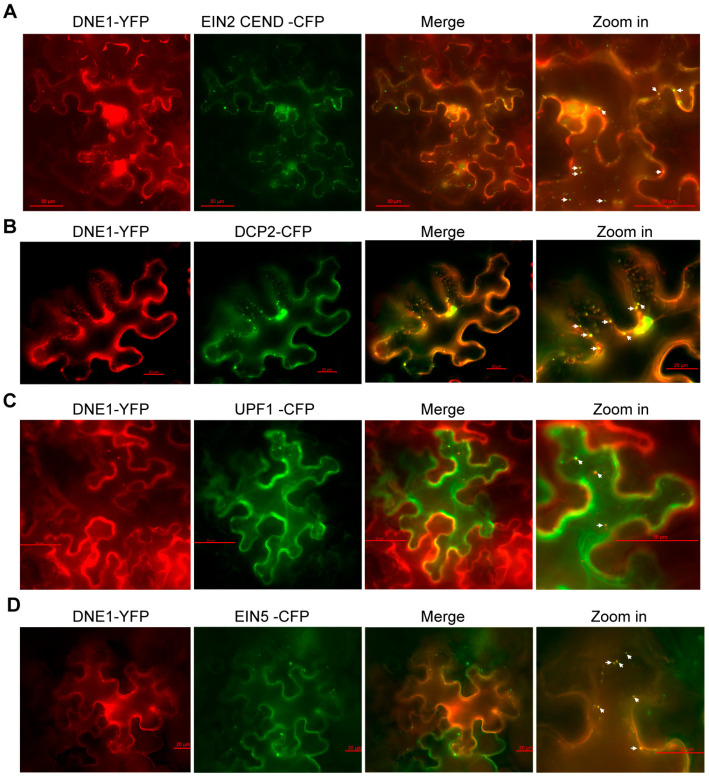
DNE1 co-localizes with EIN2 in P-body in response to ethylene. The co-localization (yellow channel) tests of DNE1 (red channel) with EIN2 CEND (**A**) and P-body markers including DCP2 (**B**), UPF1 (**C**), and EIN5 (**D**) (green channels) were performed, respectively, using Agrobacterium-mediated transient co-expression in tobacco leaves, followed by treatment with 10 μM ACC for 16 h at 2 days post-infiltration. Zoom ins on the right panel represent higher magnification images of a portion of the cell, as indicated, in which bars denote 50 μm, 20 μm, 50 μm, and 20 μm, respectively. The white arrowheads in magnification images of combined channels indicate co-localization between DNE1 and the respective proteins.

**Figure 3 ijms-25-02138-f003:**
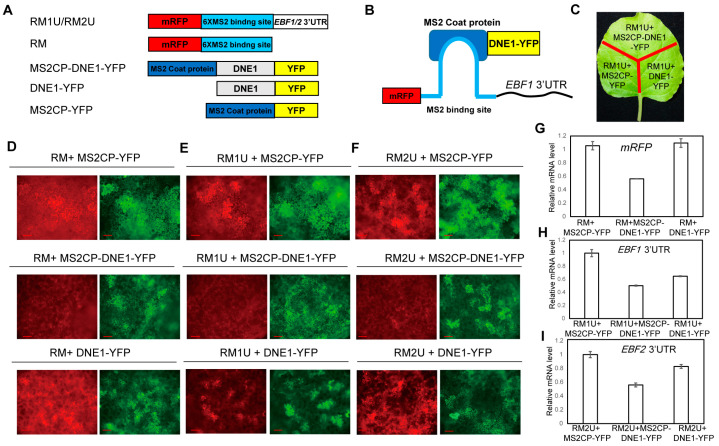
DNE1 negatively regulates reporter genes harboring *EBF1/2* 3′UTR in the tethering assay. (**A**) Schematic representation of the expression vectors used to drive the tethering assay. RM1U and RM2U are reporters, which are 6×-MS2 binding sites and *EBF1* 3′UTR or *EBF2* 3′ UTR fusion sequence linked to the stop codon of mRFP coding sequence, respectively. RM, working as the negative control, means that the 6×-MS2 binding site alone is linked to the stop codon of mRFP. MS2CP-DNE1-YFP, DNE1-YFP, and MS2CP-YFP are effectors. MS2CP-YFP stands for a MS2 coat protein-YFP fusion protein. (**B**) Schematic diagram of the MS2 system-based tethering assay, in which reporter RNA is tethered by DNE1 because the MS2 coat protein recognizes and binds to the MS2 binding site. (**C**) Schematic drawing of the tethering assay in tobacco leaves. CaMV 35S promoter-driven construct pairs were transiently co-expressed in *N. benthamiana*. Then, the florescence activities of RFP and YFP were detected for different combinations between reporters and effectors, as noted in (**A**). (**D**–**F**) Representative images of the tethering assay revealing the repressive effect of DNE1 on the expression of reporters. Three independent biological replicates were performed, and each biological replicate includes three technical repeats. Red channels shows signals of mRFP-6×MS2bs and mRFP-6×MS2bs-*EBF1/2* 3′UTR while green channels shows signals of MS2CP-DNE1-YFP, DNE1-YFP and MS2CP-YFP. Scale bars = 200 μm. (**G**–**I**) Quantitative RT–PCR revealing the relative transcript levels of *mRFP*, *EBF1* 3′UTR, and *EBF2* 3′UTR in the samples in (**D**–**F**). Total RNA was extracted from sections of tobacco leaf, as illustrated in (**C**). The *Kana* gene, located within the reporter vector, was used as the quantitative control. Triplicate biological replicate experiments, each containing three technical replicates, were performed. Values were normalized to the expression of Kana and are calculated relative to the level of the combination of reporter and MS2CP-YFP. Data are presented as mean ± SD from three biological replicates.

**Figure 4 ijms-25-02138-f004:**
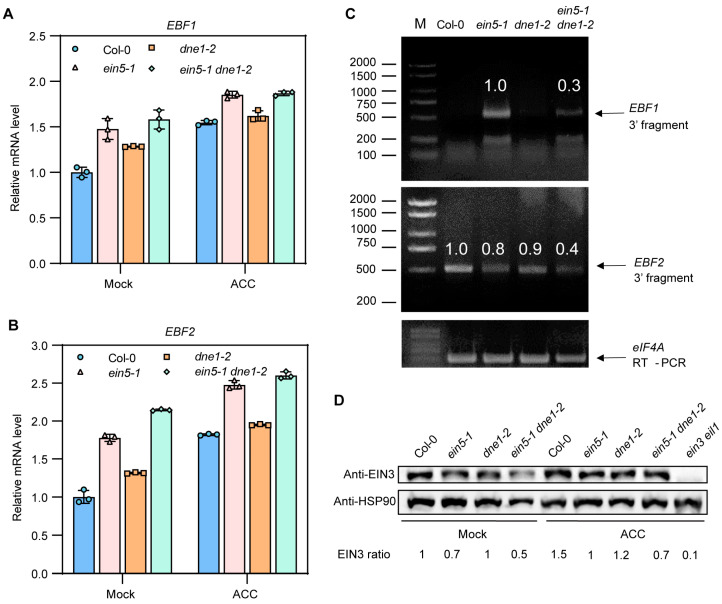
DNE1 is required for cleavage of the 3′end of *EBF1/2* mRNAs. (**A**,**B**) The relative expression levels of *EBF1/2* in seedlings in (Figure 1A) determined by qRT–PCR, in which *UBQ5* acts as an internal control. Col-0, *ein5-1*, *dne1-2*, and *ein5-1 dne1-2* were grown on MS medium supplemented with and without 10 μM ACC. The mean and SD were calculated from three independent experiments. (**C**) Using 5′RACE to detect the production of 3′ fragments of *EBF1/2* transcripts in Col-0, *ein5-1*, *dne1-2*, and *ein5-1 dne1-2* grown on MS medium treated with 10 μM ACC. (**D**) Protein levels of EIN3 in Col-0, *ein5-1*, *dne1-2*, and *ein5-1 dne1-2* grown on MS medium supplemented with and without 10 μM ACC.

**Figure 5 ijms-25-02138-f005:**
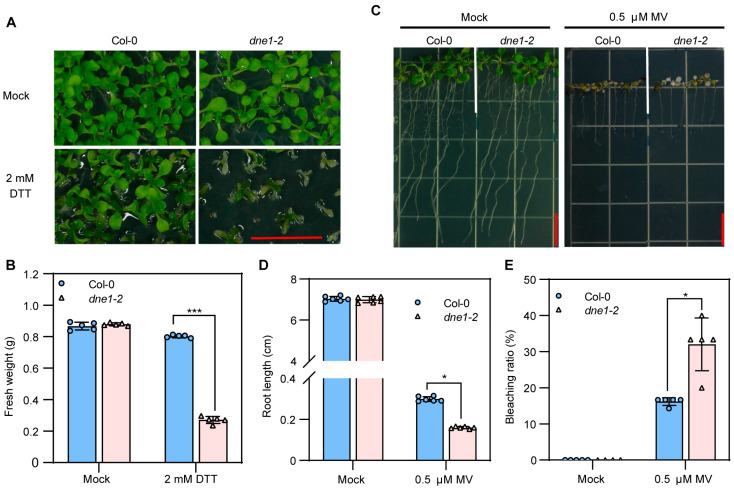
DNE1 is involved in plant responses to ER stress and oxidative stress. (**A**) Phenotypes of Col-0 and *dne1-2* grown on MS medium supplemented with and without 2 mM DTT. Scale bar = 1 cm. (**B**) Fresh weight of Col-0 and *dne1-2* in (**A**). Data are presented as mean ± SD from five biological replicates, with 10 seedlings per replicate. (**C**) Phenotype of Col-0 and *dne1-2* grown on MS medium supplemented with and without 0.5 μM MV. Scale bar = 1 cm. (**D**) Root length of Col-0 and *dne1-2* in (**D**). Data are presented as mean ± SD from six biological replicates. (**E**) Bleaching ratio of Col-0 and *dne1-2* in (**D**). Data are presented as mean ± SD from six biological replicates. For (**B**,**D**,**E**), an asterisk indicates a significant difference between different genotypes. Two-way ANOVA followed by Bonferroni’s multiple comparisons test was used. *, *** indicates a *p*-value < 0.033, or 0.001, respectively.

**Figure 6 ijms-25-02138-f006:**
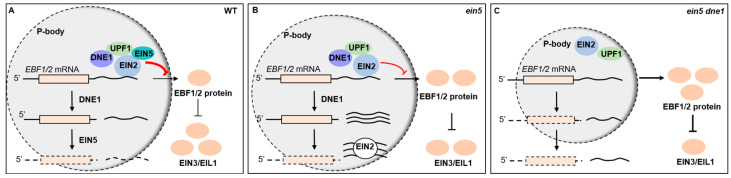
Endoribonuclease DNE1 positively regulates ethylene response in *Arabidopsis*. (**A**) The 3’UTR regions of *EBF1/2* mRNA recruit mRNA processing factors such as EIN2, EIN5, UPF1, and DNE1, and modulate EIN2-mediated translational inhibition in response to ethylene. The endonuclease DNE1, together with those RNA-associating components targeting P-bodies, recognizes and cleaves the 3′ end of *EBF1/2* mRNA to produce 3′UTR fragments, which are then subject to the exoribonuclease EIN5-dependent degradation. As a result, the translation and abundance of *EBF1/2* mRNA are significantly down-regulated, which leads to a decrease in EBF1/2 protein. So, the key transcription factors EIN3 and EIL1 accumulate to high levels. (**B**) Defects of *EIN5* promote the stabilization and accumulation of the 3′UTR fragments, which then act as a sponge to titrate EIN2 and other repressive proteins, thus alleviating the translation repression of *EBF1/2* mRNA. So, EIN3/EIL1 proteins are down-regulated along with the increase in EBF1/2 proteins. (**C**) In the *ein5-1 dne1-2* double mutant, the integrity of EIN2-targeted P-body might be compromised. Although the sponge effect is impaired by the relatively low abundance of 3′UTR fragments, *EBF1/2* mRNAs are released from EIN2-mediated translation blocking and yield highly abundant EBF1/2 proteins. Thus, EIN3/EIL1 proteins are significantly decreased, which makes a more pronounced ethylene-insensitive phenotype of the *ein5-1 dne1-2* mutant than that of *ein5-1*.

## Data Availability

Data is contained within the article and Appendix A.

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
