# Peer review of "Endoribonuclease DNE1 Promotes Ethylene Response by Modulating *EBF1/2* mRNA Processing in *Arabidopsis"

_ijms, 2024, doi:10.3390/ijms25042138_

Round 1

Reviewer 1 Report

Comments and Suggestions for Authors

The manuscript submitted to the IJMS contains interesting new data on the role of endoribonuclease DNE1 in the response of Arabidopsis plants to ethylene. The information about DNE1's new responsibilities is itself of interest. In addition, new knowledge about the most intriguing stage of ethylene signaling, namely everything related to the molecular events around EIN2, is certainly of great interest to everyone who studies the mechanisms of ethylene signal transduction in plants.

The manuscript is well written and contains reliable data.

Of course, the manuscript can be accepted for publication.

However, authors should pay attention to the following points and make corrections.

1. In the abstract (line 23) you should decipher DTT and write "oxidative stress" instead of "osmotic stress".

2. Panel B appears twice in Figure 1.

3. Lines 268–289. Written: "Meanwhile, EBF1/2 mRNA levels were significantly higher than in ein5-1 dne1-2 than ein5-1 (Figure 4A and 4B)." You should probably write: "Meanwhile, EBF1/2 mRNA levels were significantly higher in ein5-1 dne1-2 than in ein5-1 (Figure 4A and 4B)."

4. Lines 280–282. Written: "Taking the fact that the abundance of EBF2 3' fragments in Col-0, ein5-1, and dne1-2 showed comparable levels into consideration (Figure 4C), we supposed that EBF2 mRNA 3' UTR is subjected to more complicated regulation." It seems like this would be better: "Taking into consideration the fact that the abundance of EBF2 3' fragments in Col-0, ein5-1, and dne1-2 showed comparable levels (Figure 4C), we supposed that EBF2 mRNA 3' UTR is subjected to more complicated regulation."

5. Line 368. Written: "In the ein5-1 nde1-2 double mutant…" Apparently, this would be correct: "In the ein5-1 dne1-2 double mutant…"

6. Line 372. Written: "…ein5-1 nde1-2 mutant…" Apparently, this would be correct: "…ein5-1 dne1-2 mutant…"

7. Line 376. Written: "…and adaptions to environmental clues [45-47]." It seems like this would be better: "…and adaptations to environmental cues [45-47]."

8. Line 432. Written: "…nde1-2 (SALK_009495)…" You should probably write: "…dne1-2 (SALK_009495)…"

9. Line 453. Written: "4.4.5'-. RLM-RACE" You should probably write: "4.4. 5'-RLM-RACE"

10. Lines 460, 462, 463. Instead of NDE1 there should be DNE1.

Author Response

Comments:

The manuscript submitted to the IJMS contains interesting new data on the role of endoribonuclease DNE1 in the response of Arabidopsis plants to ethylene. The information about DNE1's new responsibilities is itself of interest. In addition, new knowledge about the most intriguing stage of ethylene signaling, namely everything related to the molecular events around EIN2, is certainly of great interest to everyone who studies the mechanisms of ethylene signal transduction in plants. The manuscript is well written and contains reliable data. Of course, the manuscript can be accepted for publication. However, authors should pay attention to the following points and make corrections.

Response: We greatly appreciate your encouraging and insightful review of our manuscript submitted to the International Journal of Molecular Sciences (IJMS). Your recognition of the manuscript's content and writing style is valued, and we understand the significance of providing reliable and significant data. Your positive feedback has been truly motivating for us. We are fully committed to addressing the points you have raised and are eager to implement your suggestions for further improvement. In order to ensure that our manuscript meets the high standards of IJMS, we have carefully reviewed your recommendations and made the required corrections.

  1. In the abstract (line 23) you should decipher DTT and write "oxidative stress" instead of "osmotic stress".

Response: Following your suggestion, we have now deciphered 'DTT' in the abstract to enhance clarity of understanding. Moreover, the term 'osmotic stress' has been amended to 'oxidative stress' in accordance with your recommendation. These adjustments have been implemented to ensure that the abstract accurately conveys the content and emphasis of our study.

  1. Panel B appears twice in Figure 1.

Response: We reviewed Figure 1 and fixed the duplicate in Panel B. The figure now accurately depicts the intended data, with each panel clearly and correctly labeled.

  1. Lines 268–289. Written: "Meanwhile, EBF1/2 mRNA levels were significantly higher than in ein5-1 dne1-2 than ein5-1 (Figure 4A and 4B)." You should probably write: "Meanwhile, EBF1/2 mRNA levels were significantly higher in ein5-1 dne1-2 than in ein5-1 (Figure 4A and 4B)."

Response: We appreciate the reviewer’s feedback, and have revised it accordingly.

  1. Lines 280–282. Written: "Taking the fact that the abundance of EBF2 3' fragments in Col-0, ein5-1, and dne1-2 showed comparable levels into consideration (Figure 4C), we supposed that EBF2 mRNA 3' UTR is subjected to more complicated regulation." It seems like this would be better: "Taking into consideration the fact that the abundance of EBF2 3' fragments in Col-0, ein5-1, and dne1-2 showed comparable levels (Figure 4C), we supposed that EBF2 mRNA 3' UTR is subjected to more complicated regulation."

Response: We agree with this suggestion, and have modified the sentence accordingly.

  1. Line 368. Written: "In the ein5-1 nde1-2 double mutant…" Apparently, this would be correct: "In the ein5-1 dne1-2 double mutant…"

Response: We value the reviewer’s thorough check, and have made the necessary revision.

  1. Line 372. Written: "…ein5-1 nde1-2 mutant…" Apparently, this would be correct: "…ein5-1 dne1-2 mutant…"

Response: We thank the reviewer for helpful feedback, and have corrected it accordingly.

  1. Line 376. Written: "…and adaptions to environmental clues [45-47]." It seems like this would be better: "…and adaptations to environmental cues [45-47]."

Response: We appreciate the reviewer’s suggestion, and have revised it accordingly.

  1. Line 432. Written: "…nde1-2 (SALK_009495)…" You should probably write:

"…dne1-2 (SALK_009495)…"

Response: We appreciate the reviewer’s careful check, and have revised it accordingly.

  1. Line 453. Written: "4.4.5'-. RLM-RACE" You should probably write: "4.4. 5'-RLM -RACE"

Response: We thank the reviewer for helpful suggestion, and have corrected it accordingly.

  1. Lines 460, 462, 463. Instead of NDE1 there should be DNE1.

Response: We value the reviewer’s thorough suggestion, and have made the necessary revisions.

Reviewer 2 Report

Comments and Suggestions for Authors

In the manuscript (MS) “Endoribonuclease DNE1 Promotes Ethylene Response by Modulation of EBF1/2 mRNA Processing in Arabidopsis” the authors describe the identification of DNE1 gene, that encodes a ribonuclease and was proved to be induced by ethylene treatment. The study brings novel information about the DNE1 function as ethylene modification and shed light to the efforts made to improve crops’ resiliency. The work has been carefully conducted and is mostly well written and clearly presented, and the authors were able to identify that DNE1 is responsible for EBF1/2 3’UTR cleavage, it also positively regulates ethylene response.

The article can be considered for publication after some adjustments, as follows.

Specific comments and corrections

Lines 33-37: This sentence states that researchers have extensively investigated in the past two decades but only one reference (number 2) is included to support this information. Please include more references to put the reader in perspective of these studies and what they have been doing.

Lines 60-67: Which studies? Does it belong to the authors? If so, please inform that “previous studies developed in our lab…”

Lines 81-86: Which subsequent studies? Me and the readers would like to be put in perspective… Please include references. 

Figures 1 to 6: Please short it to one single paragraph caption. 

Line 435: The correct is “Were used in Author [27]”.

Question: How were the authors able to conclude that ref [26] (line 413 and on) was not able to identify EBF1/2 just based on the following: 

“The selection criteria used by the authors was xrn4/dne1 xrn4 log2FC ≥ 2. The EBF1/2 3’ fragments did not meet this selection requirement and hence were filtered out. Additionally, the growth conditions of the plant materials were different. We used etiolated seedlings that treated by application of ethylene, whereas the before mentioned study employed green seedlings without any treatment. Whether EBF1/2 mRNAs are natural substrates of the NMD pathway is uncertain. This is because most of the substrates of DNE1 do not overlap with those of the NMD pathway.” 

For me, this seems speculation. Did you repeat the same conditions as ref 26 and then tried yours to confirm it? Even so, could we be sure? How? 

Author Response

Comments:

In the manuscript (MS) “Endoribonuclease DNE1 Promotes Ethylene Response by Modulation of EBF1/2 mRNA Processing in Arabidopsis” the authors describe the identification of DNE1 gene, that encodes a ribonuclease and was proved to be induced by ethylene treatment. The study brings novel information about the DNE1 function as ethylene modification and shed light to the efforts made to improve crops’ resiliency. The work has been carefully conducted and is mostly well written and clearly presented, and the authors were able to identify that DNE1 is responsible for EBF1/2 3’UTR cleavage, it also positively regulates ethylene response.

The article can be considered for publication after some adjustments, as follows.

Response: We are grateful for your thoughtful review and the positive remarks on our findings. It is gratifying to know that our work on the identification of the DNE1 gene and its role in ethylene response modulation has been well-received. Furthermore, we are pleased that you recognize its contribution to understanding crop resiliency. Your guidance is instrumental in assisting us to elevate the quality and impact of our manuscript. We are committed to meticulously addressing each of your points and will submit the revised manuscript for your reviewing.

Lines 33-37: This sentence states that researchers have extensively investigated in the past two decades but only one reference (number 2) is included to support this information. Please include more references to put the reader in perspective of these studies and what they have been doing.

Response: We appreciate your insightful comment on the references in lines 33-37 of our manuscript. It is crucial to furnish a comprehensive background to substantiate our assertions regarding the extensive research conducted in the past two decades. In light of your suggestion, we have enhanced this section by incorporating additional references to present a more expansive view of the numerous studies conducted in this field. The inclusion of these additional citations will enable readers to gain a more nuanced understanding of the notable advancements and diverse methodologies that have defined research in this domain over the last twenty years.

Lines 60-67: Which studies? Does it belong to the authors? If so, please inform that “previous studies developed in our lab…”

Response: Thank you for providing feedback on the section in lines 60-67 of our manuscript. We recognize the importance of clearly specifying the origin of the studies referenced in this part of our text. In response to your comment, we have revised this section to explicitly indicate whether the studies were conducted in our lab or are references to external work. This revision includes the addition of appropriate citations to enhance the research background and context, thus addressing the need for improved clarity in our manuscript.

Lines 81-86: Which subsequent studies? Me and the readers would like to be put in perspective… Please include references. 

Response: We appreciate your valuable feedback on the necessity of specific references in lines 81-86 of our manuscript. Providing well-documented sources is crucial for offering comprehensive context to our readers. As per your suggestion, we have incorporated the relevant references to the subsequent studies mentioned in this section. These additions aim to provide readers with a clearer perspective and a more detailed understanding of the research landscape related to our study.

Figures 1 to 6: Please short it to one single paragraph caption. 

Response: We value your thorough suggestion, and have made the necessary revisions. The captions for Figures 1 to 6 are now succinctly summarized in a single paragraph.

Line 435: The correct is “Were used in Author [27]”.

Response: Thank you for pointing out the necessary correction in line 435. Your attention to detail is greatly appreciated. We have updated the manuscript to reflect the source of the mutant by phrasing as “The upf1-10 mutant is a gift from Alonso-Stepanova Laboratory [18]”.

Question: How were the authors able to conclude that ref [26] (line 413 and on) was not able to identify EBF1/2 just based on the following:  

“The selection criteria used by the authors was xrn4/dne1 xrn4 log2FC ≥ 2. The EBF1/2 3’ fragments did not meet this selection requirement and hence were filtered out. Additionally, the growth conditions of the plant materials were different. We used etiolated seedlings that treated by application of ethylene, whereas the before mentioned study employed green seedlings without any treatment. Whether EBF1/2 mRNAs are natural substrates of the NMD pathway is uncertain. This is because most of the substrates of DNE1 do not overlap with those of the NMD pathway.” 

For me, this seems speculation. Did you repeat the same conditions as ref 26 and then tried yours to confirm it? Even so, could we be sure? How? 

Response: We appreciate your insightful questions. We also value your concerning about our interpretation of the findings from reference [27] (reference [33] in the revised manuscript). Conclusion in line 413 and on, aligned with the discussion, is aimed at deliberating on the results and proposing potential hypotheses, rather than asserting definitive conclusions. As a result, we have revised our word choices to prevent readers from assuming that conclusive findings have been established.

In the reference [27] (reference [33] in the revised manuscript),the authors tried to identify the targets of DNE1 using degradome sequencing. They uncovered 501 putative substrates of DNE1 in terms of transcripts with significant XRN4-sensitive 5'P sites. In detail, these substrates accumulate to higher levels in xrn4 than in dne1 xrn4 by log2 FC≥2. Given the fact that the levels of EBF1 and EBF2 mRNAs, as well as the 3’ fragments derived from them, are low under normal growth condition, as EBF1/2 are condition induced-genes, not housekeeping genes. The stringent selection criterion and the sensitivity of 5'P site capture assay together led to the exclusion of EBF1/2 mRNAs from the candidate targets of DNE1.

We conducted a preliminary bioinformatics analysis of the degradome sequencing results in reference [44], to visualize reads for EBF1/2 mRNA using integrated genome view (IGV) software (Additional figure 1). We found significant increases in reads in the 3’ regions of EBF1/2 in the xrn4 mutant compared to the wild-type (WT) (Additional figure 1), suggesting that EBF1/2 3' fragments might be XRN4 substrates, but they might not meet the criteria set for XRN4 substrates in reference [44] (see supplemental Figure S7, computational pipeline to identify decapped XRN4 substrates from polyA+ and polyA- PARE), and hence were not determined as XRN4 substrates as well as DNE1 targets. However, based on previous research findings in reference [25] (Figure 2, the half-life of the 3’ fragment of At2g25490 (EBF1) is significantly extended in the xrn4 mutant) and our own 5’RACE results, we observed significant accumulation of EBF1/2 3' fragments in the xrn4 mutant, suggesting that these fragments are substrates of XRN4.

The findings from references [25], [31-33], and [44] significantly advance our understanding of RNA decay and its regulation. The interpretation of our result is greatly influenced by these studies. Authors corresponding to these references are skilled at performing degradome sequencing, which we are not. So, we did not repeat the sequencing results from reference [27] (reference [33] in the revised manuscript). We believe it is reasonable for experimental results and high-throughput sequencing data to match imperfectly. Moreover, the stage and growth conditions of the plant materials in our study differ significantly from those in reference [27] (reference [33] in the revised manuscript) and reference [44], which could impact the accumulation of EBF1/2 mRNAs. In this manuscript, we did not employ the growth conditions stated by the authors in references [27] (reference [33] in the revised manuscript) to study the effect of DNE1 on EBF1/2 mRNAs, as our study focus on the influence of DNE1 on EBF1/2 in etiolated seedlings under the treatment of ethylene. This is more relevant to the ethylene signaling pathway that we aim to investigate. Our findings suggest that researchers could improve their search for DNE1 targets by broadening the growth conditions and data selection criteria.

Additional figure 1. Integrated Genome View (IGV) of EBF1/2 in polyA+ and polyA-  PARE libraries.

Reviewer 3 Report

Comments and Suggestions for Authors

Re: ijms 2836030

Yan et al submitted manuscript entitled” Endoribonuclease DNE1 Promotes Ethylene Response by Modulation of EBF1/2 mRNA Processing in Arabidopsis” for consideration of publication in International Journal of Molecular Science, ijms

Author reported here a study dealing with mechanisms modulating the processing of EBF1/2 mRNAs, they identified gene DNE1, which encodes endoribonuclease and is induced by ethylene treatment, as a positive regulator of ethylene. Author showed DNE1 colocalizes with EIN2, the core factor manipulating the translation of EBF1/2, and targets to the P-body in response to ethylene as well.   They approved that NE1 negatively regulates the abundance of EBF1/2 mRNAs by recognizing and cleaving their 3’UTRs, which leading to represses their translation. This study proposed the essential role of DNE1 as a modulator of ethylene signaling through regulation of EBF1/2 mRNA processing and provide data to support the conclusion. 

The research was well designed and executed, the presentation of results, M&M, discussion is sound. 

Howeer, there is some issues should be addressed.  

1.     In Figure 1. DNE1 positively regulates ethylene response in Arabidopsis. There is mislabelling explaining figure, (C-D) was labelled as (B, C). 

2.     Related to EBF1/EBF2 and modulates the ethylene response to abiotic stress, research conducted by Hao et al should be cited and discussed with current report ( Hao D, Jin L, Wen X, Yu F, Xie Q, Guo H. The RING E3 ligase SDIR1 destabilizes EBF1/EBF2 and modulates the ethylene response to ambient temperature fluctuations in Arabidopsis. Proc Natl Acad Sci U S A. 2021 Feb 9;118(6):e2024592118. doi: 10.1073/pnas.2024592118. PMID: 33526703; PMCID: PMC8017691.). 

3.     In Figure 6, it would be better if the models shown in colors.  

I think it is suitable for publication with ijms after careful revision

Author Response

Comments:

Yan et al submitted manuscript entitledEndoribonuclease DNE1 Promotes Ethylene Response by Modulation of EBF1/2 mRNA Processing in Arabidopsis” for consideration of publication in International Journal of Molecular Science, ijms. Author reported here a study dealing with mechanisms modulating the processing of EBF1/2 mRNAs, they identified gene DNE1, which encodes endoribonuclease and is induced by ethylene treatment, as a positive regulator of ethylene. Author showed DNE1 colocalizes with EIN2, the core factor manipulating the translation of EBF1/2, and targets to the P-body in response to ethylene as well.   They approved that NE1 negatively regulates the abundance of EBF1/2 mRNAs by recognizing and cleaving their 3’UTRs, which leading to represses their translation. This study proposed the essential role of DNE1 as a modulator of ethylene signaling through regulation of EBF1/2 mRNA processing and provide data to support the conclusion.  The research was well designed and executed, the presentation of results, M&M, discussion is sound. 

Response: We sincerely appreciate the time and effort you have invested in reviewing our manuscript. Your comprehensive summary and understanding of our study's key findings are truly valued by us. We believe that your insights not only affirm the validity of our research but also enhance the overall clarity and impact of our findings. Your contribution has been instrumental in refining the quality of our work, and we are grateful for your support in advancing the scholarly conversation in this field.

  1. In Figure 1. DNE1 positively regulates ethylene response in Arabidopsis. There is mislabelling explaining figure, (C-D) was labelled as (B, C).

Response: We appreciate the comments from the reviewer. We revised the Figure 1 accordingly.

  1. Related to EBF1/EBF2 and modulates the ethylene response to abiotic stress, research conducted by Hao et al should be cited and discussed with current report ( Hao D, Jin L, Wen X, Yu F, Xie Q, Guo H. The RING E3 ligase SDIR1 destabilizes EBF1/EBF2 and modulates the ethylene response to ambient temperature fluctuations in Arabidopsis. Proc Natl Acad Sci U S A. 2021 Feb 9;118(6):e2024592118. doi: 10.1073/pnas.2024592118. PMID: 33526703; PMCID: PMC8017691.).

Response: We have taken your suggestion into account and have incorporated a citation and thorough discussion of the research conducted by Hao et al. on the modulation of the ethylene response to abiotic stress by EBF1/EBF2 proteins. This addition can be found in line 72-74 and discussion section, in which we have connected their findings with our current report, providing a coherent and comprehensive examination of the research landscape in this field. Thank you for bringing this valuable contribution to our attention.

  1. In Figure 6, it would be better if the models shown in colors.

Response: Thanks for the reviewer’s helpful suggestion. We have colored the model in figure 6, which we believe will significantly improve the clarity of our manuscript.